# Low capacity for molecular detection of *Alphaviruses* other than Chikungunya virus in 23 European laboratories, March 2022

Laura Pezzi[1,2⊙], Ramona Moegling[3⊙], Cécile Baronti[1], Kamelia R. Stanoeva[3], Lance D. Presser[3], Pauline Jourdan[1], Nazli Ayhan[1,2], Willem M.R. van den Akker[3], Stephan Zientara[4], Céline M. Gossner[5], Rémi N. Charrel[1,6,7‡], Chantal B.E.M. Reusken[3‡*], on behalf of EVD-LabNet[¶]

**1** Unité des Virus Emergents (UVE: Aix-Marseille Univ, Universita di Corsica, IRD 190, Inserm, IRBA), Corsica, France, **2** National Reference Center for Arboviruses, National Institute of Health and Medical Research (Inserm) and French Armed Forces Biomedical Research Institute (IRBA), Marseille, France, **3** Centre for Infectious Disease Control, National Institute for Public Health and the Environment (RIVM), Bilthoven, The Netherlands, **4** Agency for Food, Environmental and Occupational Health and Safety (ANSES), Maison Alfort, France, **5** Disease Programme Unit, European Centre for Disease Prevention and Control (ECDC), Solna, Sweden, **6** Laboratoire Infections Virales Aigues et Tropicales, APHM Hôpitaux Universitaires de Marseille, Marseille, France, **7** LE Service de Prévention du Risque Infectieux (LESPRI), CLIN AP-HM Hôpitaux Universitaires de Marseille, Marseille, France

¶ Contributing members of EVD-LabNet are provided in the Acknowledgments.
⊙ These authors contributed equally to this work.
‡ RNC and CBEMR also contributed equally to this work.
* chantal.reusken@rivm.nl

## Abstract

Alphaviruses comprise over 30 identified species spread worldwide and carry a large global health burden. With vector expansion occurring in and around Europe, it is anticipated this burden will increase. Therefore, regular assessment of the diagnostic capabilities in Europe is important, e.g., by conducting external quality assessments (EQAs). Here we evaluated molecular detection of alphaviruses in expert European laboratories by conducting an EQA in March 2022. Molecular panels included 15 samples: nine alphaviruses, Barmah Forest virus (BFV), chikungunya virus (CHIKV), Eastern equine encephalitis virus (EEEV), Mayaro virus (MAYV), o'nyong-nyong virus (ONNV), Ross River virus (RRV), Sindbis virus (SINV), Venezuelan equine encephalitis virus (VEEV), and Western equine encephalitis virus (WEEV) and four negative control samples. Alphavirus detection was assessed among 23 laboratories in 16 European countries. Adequate capabilities were lacking for several viruses, and approximately half of the laboratories (11/23) relied on pan-alphavirus assays with varying sensitivity and specificity. Only 46% of laboratories characterized all EQA samples correctly. Correct result rates were >90% for CHIKV, RRV and SINV, but laboratories lacked specificity for ONNV and MAYV and sensitivity for VEEV, BFV, and EEEV. Only two alphaviruses causing human disease circulate or have circulated in Europe, CHIKV and SINV. Molecular detection was satisfactory with both CHIKV and SINV, but <50% correct for the entire alphaviruses panel. With continued imported cases, and a growing global concern about climate change and vector

**Data availability statement:** All relevant data are within the paper and its Supporting Information files.

**Funding:** This work was financed by the European Centre for Disease Prevention and Control (ECDC) under specific contract No 1 ECD.11886 ID 23729 implementing Framework contract No ECDC/2020/010. For providing access to the virus strains used, we thank the EVA-GLOBAL consortium (funded by the European Union's Horizon 2020 research and innovation programme under grant agreement No. 871029). Except VEEV, WEEV and EEEV, the material was provided by the European virus archive-Marseille (EVAM) under the label technological platforms of Aix-Marseille University. CBEMR and RNC were both coordinators or work package leaders for ECDC and EU Horizon funding.

**Competing interests:** The authors have declared that no competing interests exist.

expansion, focus on progress toward rapid, accurate alphavirus diagnostics in Europe is recommended, as well as regular EQAs to monitor quality.

## Introduction

Alphaviruses are positive-sense RNA viruses that belong to the *Alphavirus* genus within the *Togaviridae* family [1]. Alphaviruses are transmitted by arthropods (mainly mosquitoes of the genera *Culex, Culiseta, Aedes, Coquillettidia,* and *Haemogogus*), between vertebrate species. With over 30 known species, alphaviruses cause considerable disease burden for humans, with regional and seasonal variances. Several alphaviruses have caused large outbreaks or have potential for future outbreaks and/or geographic expansion [2–6]. Such dissemination was exemplified in 2013–2014 when CHIKV entered and spread in North/South America [7,8].

In Europe, only one alphavirus of public health importance has endemic presence, i.e., SINV. SINV causes localized human outbreaks in northern Europe [6,9]. The main mosquito vector for SINV transmission from birds to humans, *Culex pipiens* is established across large areas of Europe and North Africa [10]. SINV RNA and/or antibodies to SINV have been identified in mosquitoes and wildlife in various countries across Europe [11–15]. It is assumed that this pathogen is under-diagnosed [16,17].

Autochthonous transmission of CHIKV has been reported in France and Italy between 2007–2017, with hundreds of cases in Italy in 2007 and 2017 [18–20]. Local transmission in the EU/EEA follows introduction of the virus to *Aedes albopictus* infested areas by viraemic travellers returning from CHIKV endemic regions in (sub)tropical regions, including some EU outermost regions (OR) and/or EU member state overseas countries and territories (OCT).

*Ae. albopictus*, identified as the mosquito species responsible for local CHIKV transmission in Europe, is established in a large part of southern and central Europe and establishment has reached as far north as Brandenburg state in Germany with sightings of the species as far north as Sweden [21]. It is predicted that future climate trends will make Europe increasingly suitable for *Ae. albopictus* establishment in the next decades, highlighting an increased risk for CHIKV local transmission [21,22]. Other native or invasive mosquito species established in Europe have proved to be susceptible to CHIKV infection (*Ae. japonicus* [23]*, Ae. vexans* [24]) or showed capacity of transmitting CHIKV (*Ae. koreicus* [25,26]) in experimental laboratory settings, but their possible role as secondary vectors in CHIKV epidemiology in Europe is unknown.

Other alphaviruses for which EU travellers are at risk across the world are RRV (Australia and Pacific ocean area), BFV (Australia), WEEV (the America's), VEEV (Central and South America), EEEV (the America's), ONNV (Africa), and MAYV (Central/South America and Caribbean) [27–29]. RRV, ONNV, MAYV and EEEV have been imported by travellers into Europe [30–34]. However, contrary to CHIKV, the import of these viruses to the EU/EEA by returning travellers is not expected to cause autochthonous cases as humans are considered dead-end hosts for ONNV, MAYV, EEEV while for RRV the role of humans as reservoir for mosquito-borne infections seems to depend on very specific circumstances [27,28,35]. For WEEV, VEEV and BFV, infections in travellers have so far not been reported. Based on current knowledge, such imports are not expected to lead to autochthonous mosquito-borne transmission [36,37]. However, live-animal trade might be a source for introduction and transmission of some of these alphaviruses, potentially leading to ongoing local transmission among animal reservoirs and spill-over to humans [38,39]. For all these viruses except CHIKV, it remains to be seen whether competent vectors are

present in Europe and whether the vector capacity will be such that those competent species will become ecologically significant in local transmission of these viruses [40–42]. Nevertheless, the reliable detection of these viruses provides important information on specific alphavirus epidemiology in the countries where the infection was acquired, enables correct patient management and supports risk assessment for public health threats in the EU and in EU-travellers in particular [30–34].

One of the main components of outbreak and pandemic preparedness and response is the ability to diagnose or rule out infections accurately and timely for individual patient management, surveillance, and control purposes [43–45]. Diagnostic methods need to be established, maintained, and evaluated independently and regularly, e.g., by conducting external quality assessments (EQAs) through proficiency testing [46–50]. Given that CHIKV will continue to be an emerging threat for European countries with established populations of *Ae. albopictus* and *Ae. aegypti* [21,51], and travellers will continue to return with alphavirus infections, the Emerging Viral Diseases-Expert Laboratory Network (EVD-LabNet) funded by the European Centre for Disease Prevention and Control (ECDC) [52] conducted an EQA for molecular detection of alphaviruses among its members in 2022.

## Results

### Participation and testing capacity

In total 28 laboratories registered and received EQA panels, of which four did not submit results and one laboratory was excluded from the analysis because their submitted results were incomplete. One laboratory notified they would not be submitting results due to shortage of test kits. Another laboratory notified of shortage of staff to complete the EQA. Overall, 23 laboratories from 15 EU/EEA countries and one EU pre-accession country participated (Fig 1).

Twelve laboratories in eight countries indicated capacity to detect and identify all EQA panel samples based on their tests. Four laboratories in four countries had only capacity to detect CHIKV RNA among the participants, while four laboratories in four other countries had different virus species combinations in the capacity among their participants varying from two virus species to five virus species (S1 Table).

The median number of test methods that participant laboratories used to analyse the EQA panels was three, with the maximum number of test methods performed by a single laboratory being nine (Table 1). The testing capacity for each specific alphavirus varied significantly: while all 23 laboratories performed at least one CHIKV-specific test method, only 13 of the 23 laboratories performed at least one BFV-specific test method (Table 1). Overall, ONNV had the second highest capacity with 17 laboratories performing specific testing (species-specific RT-PCR and/or pan-alphavirus RT-PCR in combination with sequencing) while MAYV had the second highest capacity when looking at species-specific RT-PCR testing alone (Table 1).

Two laboratories exclusively tested the whole EQA panel by performing a pan-alphavirus assay combined with sequencing (Table 1). The remaining twenty-one laboratories used specific RT-PCR assays for (a subset of) the alphaviruses; 12 laboratories only relied on species-specific RT-PCR test methods, of which only one laboratory was able to discriminate between all nine alphaviruses and of which five laboratories were solely able to determine the presence or absence of CHIKV in the EQA panel. One laboratory used a Whole Transcriptome Amplification (WTA) detection array that should be able to differentiate between the nine different alphaviruses. Eight laboratories used a combination of species-specific RT-PCR test methods and a pan-alphavirus assay combined with sequencing. One laboratory performed specific RT-PCR test methods and additionally High-Throughput Sequencing (HTS) (Table 1).

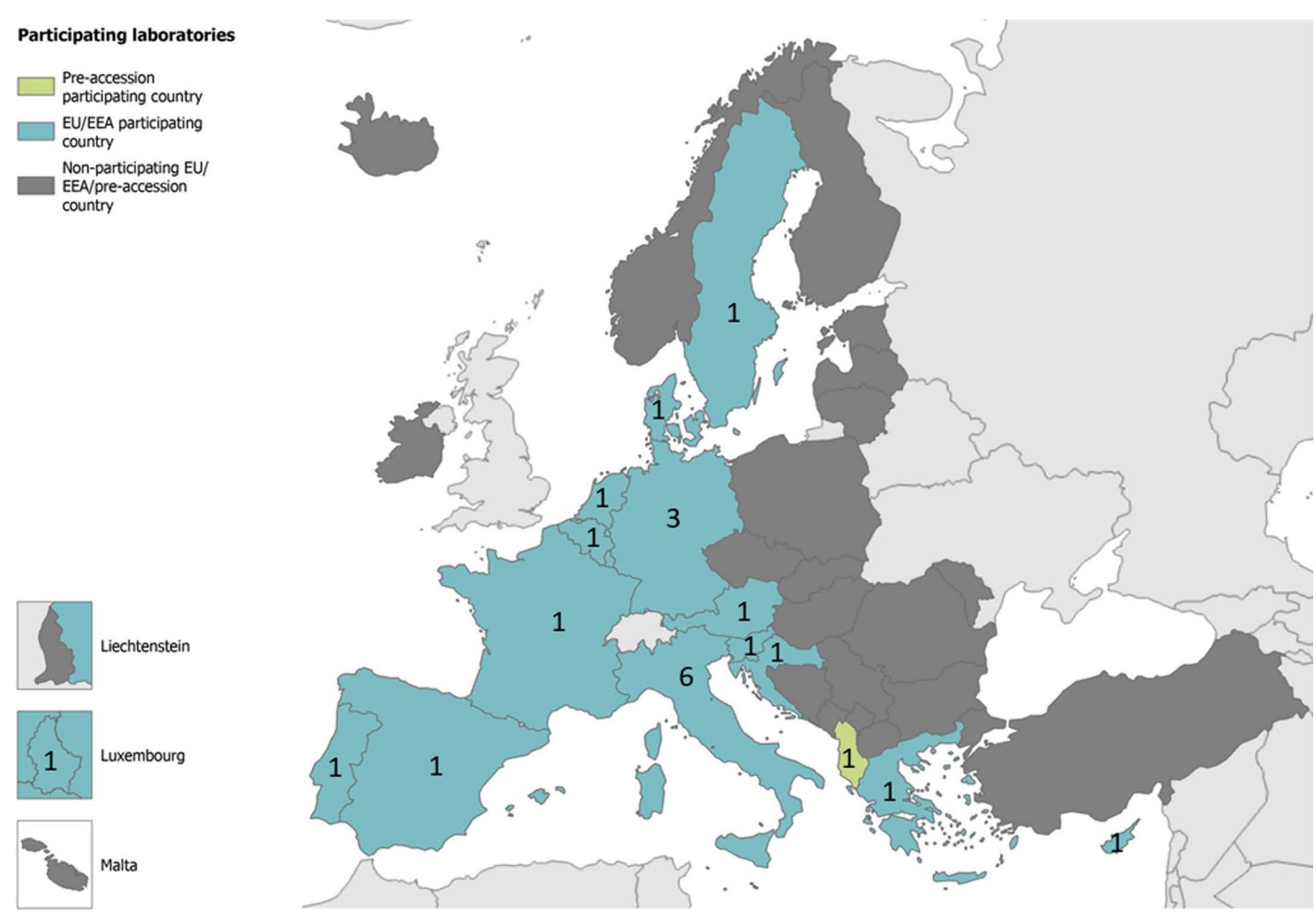

**Fig 1. Map of participating countries and number of laboratories per country in the 2022 EVD-LabNet EU/EEA External Quality Assessment for molecular detection of alphaviruses.**

Fifteen laboratories used automated extraction of nucleic acids, while eight used manual extraction.

### Diagnostic interpretation

When looking at the overall diagnostic interpretation of each of the 23 participants for the 15 EQA panel entries and considering their self-reported testing capacity, 11 of 23 laboratories had a correct final interpretation of all 15 EQA panel entries (Fig 2A). Of these 11 laboratories, six laboratories correctly interpreted the complete EQA panel at the species-specific level, while five laboratories had species-specific test methods for either 5, 4 or 1 of the nine alphaviruses and correctly called all of them according to their capacity (Table 3, Fig 2A).

Twelve of 23 laboratories had at least one error in their final interpretation of the 15 panel entries based on their indicated capacities (Fig 2A). Nine of these 12 laboratories had a single sample incorrect (seven times *'false positive'* and two times *'false negative'*, Fig 2A). Of these nine laboratories, four reported to be able to discriminate between all nine alphaviruses in the panel while five had species-specific test methods for either 7, 5, 4 or 1 of the nine

**Table 1. Alphavirus testing capacity as indicated by participating laboratories in the 2022 EVD-LabNet External Quality Assessment of molecular testing for nine human alphaviruses.**

| lab code | # test methods ↓ used/lab | Alphavirus species detectable with ≥ 1 assays | | | | | | | | | Key |
| --- | --- | --- | --- | --- | --- | --- | --- | --- | --- | --- | --- |
| | | CHIKV | ONNV | RRV | SINV | EEEV | VEEV | WEEV | MAYV | BFV | |
| A | 1 | | | | | | | | | | pan-alphavirus + seq |
| B | 3 | | | | | | | | | | specific RT-PCR test method(s) or WTA |
| C | 3 | | | | | | | | | | pan-alphavirus & pan-alphavirus + seq |
| D | 4 | | | | | | | | | | specific RT-PCR(s) & pan-alphavirus + seq / HTS |
| E | 3 | | | | | | | | | | test methods absent |
| F | 1 | | | | | | | | | | |
| H | 1 | | | | | | | | | | |
| G | 5 | | | | | | | | | | |
| I | 3 | | | | | | | | | | |
| J | 5 | | | | | | | | | | |
| K | 7 | | | | | | | | | | |
| L | 9 | | | | | | | | | | |
| M | 8 | | | | | | | | | | |
| N | 7 | | | | | | | | | | |
| O | 2 | | | | | | | | | | |
| P | 2 | | | | | | | | | | |
| Q | 7 | | | | | | | | | | |
| R | 2 | | | | | | | | | | |
| S | 1 | | | | | | | | | | |
| T | 2 | | | | | | | | | | |
| U | 2 | | | | | | | | | | |
| W | 1 | | | | | | | | | | |
| X | 1 | | | | | | | | | | |
| # labs test method present | → | 23/23 | 17/23 | 16/23 | 15/23 | 15/23 | 15/23 | 15/23 | 14/23 | 13/23 | |
| % | → | 100 | 74 | 70 | 65 | 65 | 65 | 65 | 61 | 57 | |

Each participating laboratory has a unique identifier laboratory code (A-X). CHIKV: chikungunya virus (all three lineages combined); SINV: Sindbis virus; BFV: Barmah Forest virus; RRV: Ross River virus; ONNV: o'nyong-nyong virus, MAYV: Mayaro virus, EEEV: Eastern equine encephalitis virus; VEEV: Venezuelan equine encephalitis virus; WEEV: Western equine encephalitis virus; sequencing (seq); Reverse transcription polymerase chain reaction (RT-PCR); Whole Transcriptome Amplification (WTA); High-Throughput Sequencing (HTS).

alphaviruses. The remaining four of the 12 laboratories mis-interpreted three or more EQA samples (Fig 2A).

When looking at the overall diagnostic interpretation per virus species/lineage by the 23 participants based on their self-reported capacity, six of the 11 viral RNA containing panel entries were accurately interpreted by ≥95% of the laboratories, i.e., CHIKV-ECSA, CHIKV-Asian, CHIKV-WA, RRV-RNA, SINV and WEEV RNA (Fig 2B). For the two panel entries containing either ONNV or MAYV RNA, the main problem for the correct final diagnostic interpretation by laboratories was a lack of specificity (Fig 2B): five laboratories incorrectly identified the ONNV-RNA entry as CHIKV-RNA positive, while two laboratories identified

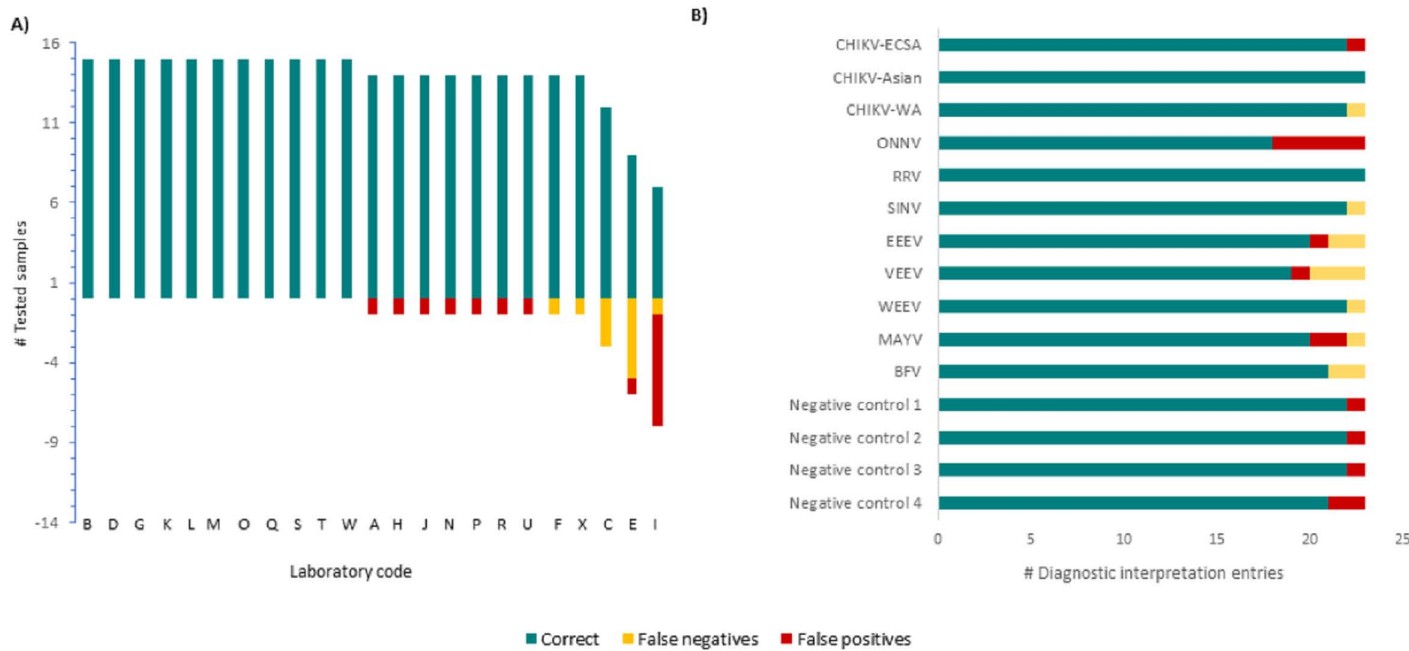

**Fig 2. Overview of final diagnostic conclusions assessed per self-reported capacities in the 2022 EVD-LabNet External Quality Assessment of molecular testing for alphaviruses.**

the MAYV-RNA entry as ONNV-RNA positive (S1 Table). In contrast, for the VEEV, EEEV and BFV- RNA containing panel entries a lack of sensitivity was observed, with two or three laboratories incorrectly identifying these samples as negative (Fig 2B). The four negative controls were correctly identified as negative for alphavirus RNA by most laboratories (20 of 23) (Fig 2B). The false positive results that were reported, included BFV, CHIKV or an un-specified alphavirus.

Final diagnostic conclusions for 15 panel entries by 23 laboratories (A) and by specific EQA-panel entry for 23 result entries (one final conclusion per participant) (B). In (A) the x-axis indicates correctly identified panel entries in the plus scale and the wrongly identified entries in the minus scale with a maximum of 15 panel entries with a final diagnostic conclu-sion per laboratory according to their self-reported capacities (Table 1). Unique laboratory codes are indicated for each set of diagnostic conclusions. CHIKV: chikungunya virus; SINV: Sindbis virus; BFV: Barmah Forest virus; RRV: Ross River virus; ONNV: o'nyong-nyong virus, MAYV: Mayaro virus, EEEV: Eastern equine encephalitis virus; VEEV: Venezuelan equine encephalitis virus; WEEV: Western equine encephalitis virus.

### Assay performances per alphavirus species

**Chikungunya virus.** All 23 laboratories indicated to have capacity for specific testing for CHIKV, using a great variety of assays. In total there were 27 different methods used, of which 26 should detect CHIKV at the species level. These included 18 species-specific RT-PCR methods, six pan-alphavirus RT-PCR tests in combination with sequencing, one WTA and one HTS. Furthermore, there was one pan-alphavirus RT-PCR test at genus level (Table 2 and S2 Table). There were commercial and in-house assays, real-time and conventional RT-PCRs (Table 2 and S2 Table). The three main CHIKV genetic lineages were represented in the EQA panel. Overall, the 23 participants submitted 38 result entries for the Asian and

**Table 2. Performance of all assays used in the 2022 EVD-LabNet EQA on molecular detection of alphaviruses. Incorrect results can be either false negative, false positive for a different alpha virus species, false positive for a negative sample or inconclusive.**

| Assay design | Method | Result level | RRV | SINV | WEEV | VEEV | EEEV | BFV | MAYV | ONNV | CHIKV-Asian | CHIKV-West African | CHIKV-ECSA | Negative samples | False positive (FP) | False negative (FN) | Inconclusive | Correct entries, n (%) |
|---|---|---|---|---|---|---|---|---|---|---|---|---|---|---|---|---|---|---|
| Pan-alphavirus RT-PCR | In-house[a] | Genus | 2/2 | 2/2 | 2/2 | 1/2 | 2/2 | 2/2 | 2/2 | 2/2 | 2/2 | 2/2 | 2/2 | 5/8 | 3 | 1 | – | 26/30 (87%) |
| | In-house followed by sequencing[b] | Species | 11/11 | 10/13 | 9/13 | 6/13 | 6/13 | 8/11 | 8/11 | 10/11 | 11/11 | 11/11 | 10/11 | 47/52 | 9 | 23 | 2 | 147/181 (81%) |
| WTA | In-house | | 1/1 | 1/1 | 1/1 | 1/1 | 1/1 | 1/1 | 1/1 | 1/1 | 1/1 | 1/1 | 1/1 | 3/4 | 1 | – | – | 14/15 (93%) |
| HTS | N/A | | 1/1 | 1/1 | 1/1 | 1/1 | 1/1 | 1/1 | 1/1 | 1/1 | 1/1 | 1/1 | 1/1 | 4/4 | – | – | – | 15/15 (100%) |
| RRV-specific RT-PCR | In-house | Species | 6/6 | 6/6 | 6/6 | 6/6 | 6/6 | 6/6 | 5/6 | 6/6 | 6/6 | 6/6 | 6/6 | 24/24 | – | – | 1 | 89/90 (99%) |
| SINV-specific RT-PCR | In-house | Species | 6/6 | 5/6 | 6/6 | 5/6 | 6/6 | 6/6 | 6/6 | 6/6 | 6/6 | 6/6 | 6/6 | 24/24 | 1 | 1 | – | 88/90 (98%) |
| WEEV-specific RT-PCR | In-house | Species | 4/4 | 4/4 | 4/4 | 4/4 | 4/4 | 4/4 | 4/4 | 4/4 | 4/4 | 4/4 | 4/4 | 16/16 | – | – | – | 60/60 (100%) |
| WEEV-specific RT-PCR | Commercial | | 1/1 | 1/1 | 1/1 | 1/1 | 1/1 | 1/1 | 1/1 | 1/1 | 1/1 | 1/1 | 1/1 | 4/4 | – | – | – | 15/15 (100%) |
| VEEV-specific RT-PCR | In-house | Species | 3/3 | 3/3 | 3/3 | 3/3 | 3/3 | 3/3 | 3/3 | 3/3 | 3/3 | 3/3 | 3/3 | 12/12 | – | – | – | 45/45 (100%) |
| VEEV-specific RT-PCR | Commercial | | 1/1 | 1/1 | 1/1 | 1/1 | 1/1 | 1/1 | 1/1 | 1/1 | 1/1 | 1/1 | 1/1 | 4/4 | – | – | – | 15/15 (100%) |
| EEEV-specific RT-PCR | In-house | Species | 5/5 | 5/5 | 5/5 | 5/5 | 5/5 | 5/5 | 5/5 | 5/5 | 5/5 | 5/5 | 5/5 | 20/20 | – | – | – | 75/75 (100%) |
| EEEV-specific RT-PCR | Commercial | | 1/1 | 1/1 | 1/1 | 1/1 | 1/1 | 1/1 | 1/1 | 1/1 | 1/1 | 1/1 | 1/1 | 4/4 | – | – | – | 15/15 (100%) |
| BFV-specific RT-PCR | In-house | Species | 3/3 | 3/3 | 3/3 | 3/3 | 3/3 | 3/3 | 3/3 | 3/3 | 3/3 | 3/3 | 3/3 | 12/12 | – | – | – | 45/45 (100%) |
| MAYV-specific RT-PCR | In-house | Species | 8/8 | 8/8 | 8/8 | 8/8 | 8/8 | 8/8 | 8/8 | 8/8 | 8/8 | 7/8 | 8/8 | 31/32 | – | 1 | 1 | 118/120 (98%) |
| ONNV-specific PCR | In-house | Species | 8/8 | 8/8 | 8/8 | 8/8 | 8/8 | 8/8 | 8/8 | 6/8 | 8/8 | 8/8 | 8/8 | 32/32 | – | 2 | – | 118/120 (98%) |
| CHIKV-specific RT-PCR | In-house | Species | 16/17 | 16/17 | 16/17 | 16/17 | 16/17 | 16/17 | 16/17 | 14/17 | 17/17 | 16/17 | 15/17 | 64/68 | 3 | 3 | 11 | 238/255 (93%) |
| CHIKV-specific RT-PCR | Commercial[c] | | 6/6 | 7/7 | 7/7 | 7/7 | 6/7 | 6/6 | 6/6 | 3/6 | 5/6 | 7/7 | 5/6 | 27/28 | 4 | 2 | 1 | 92/99 (93%) |

[a]Both laboratories that used this method to test the samples on genus level, also used additional other methods that could differentiate between the different samples.

[b]Although these methods can be used to detect all alpha samples of this EQA, one laboratory used these methods only to analyze a subset of the EQA samples, as indicated by the total numbers per samples.

[c]One laboratory used this method to analyze only a subset of the EQA samples, as indicated by the total number of tests per sample. They did not explain this deviation.

the ECSA sample each, and 39 result entries for the WA sample. The discrepancy is due to one laboratory not analysing all the EQA panel samples with their CHIKV-specific assay. Regardless of design, most assays were sensitive for CHIKV detection, with one false-negative result for each the CHIKV-Asian and CHIKV-WA sample and four false negative results for the CHIKV-ECSA sample. These four false negative results were obtained with four different CHIKV-specific methods (Table 2 and S2 Table). In contrast to sensitivity, recurrent problems of specificity were observed: a) the ONNV RNA-positive sample was detected as CHIKV RNA-positive by six laboratories, using five different methods; b) one CHIKV-specific RT-PCR gave an inconclusive result for the RRV-positive sample; c) inconclusive results were reported for five negative control samples using two different CHIKV-specific RT-PCR tests, as well as one false CHIKV RNA-positive result for a negative control sample. Overall, the proportion of correct entries for the CHIKV-RNA positive samples ranged from 90% (n = 35/39; ECSA sample) to 97% (n = 38/39; Asian sample) and 98% (n = 39/40; WA sample).

**Sindbis virus.** Overall, 16 laboratories in nine countries tested 15 different methods that should detect the SINV containing sample, of which 14 methods at the virus species level. Of these, five methods were SINV-specific RT-PCRs, seven conventional pan-alphavirus RT-PCRs in combination with sequencing, one WTA and one HTS. In total, the 15 different methods yielded 23 result entries for the SINV-RNA panel (Table 2 and S2 Table).

Thirteen results were submitted for the SINV sample using a pan-alphavirus RT-PCR test in combination with sequencing, of which three provided a false-negative result, showing a lack of sensitivity for SINV detection and two provided a false positive result (falsely categorizing either CHIKV or VEEV as SINV-positive), indicative for possible specificity issues. Among the six results for the SINV sample obtained with the five SINV-specific RT-PCRs, there was one false positive (detection of the VEEV-positive sample as SINV-positive) and one false negative result. Since the same laboratory detected the VEEV-positive sample as SINV-positive by both a pan-alphavirus in combination with sequencing assay and a SINV species-specific RT-qPCR assay, it is possible that the specificity issue of this laboratory could be explained by either cross-contamination or a mix up the results during the reporting of the EQA.

**Ross river virus.** Overall, 17 laboratories in 10 countries tested the proficiency panel with 15 different methods to detect RRV positive samples, of which one method could only identify to genus level and 14 methods could identify to the RRV-species-level. These 14 methods included six different in-house developed RRV-specific RT-PCRs (by five laboratories), six pan-alphavirus RT-PCRs in combination with sequencing, one WTA and one HTS. In total, the 15 different methods yielded 21 result entries for the RRV panel entry by the 17 laboratories (Table 2 and S2 Table). All applied methods scored 100% for the RRV sample. While no false-positive results were reported there was an inconclusive call on the MAYV sample.

**Western equine encephalitis virus.** Overall, 16 laboratories in nine countries tested the proficiency panel with 14 different methods to detect WEEV-positive samples, of which one method could only identify to genus level and 13 methods could identify to the WEEV species-level. These 13 methods included four different WEEV-specific RT-PCRs, seven pan-alphavirus RT-PCRs in combination with sequencing, one WTA and one HTS. In total the 14 different methods yielded 22 result entries for the WEEV sample by 16 laboratories (Table 2 and S2 Table).

All WEEV-specific methods provided correct results. All results of the WEEV-specific RT-PCRs were correct, indicating reliable specificity and sensitivity. However, pan-alphavirus assays in combination with sequencing showed good specificity (no false positive results) but low sensitivity, i.e., four of these entries were false negative. (Table 2 and S2 Table).

**Venezuelan equine encephalitis virus.** Overall, 16 laboratories in nine countries tested with 13 different methods to detect VEEV-positive samples of which 12 at the species level. Of these, three methods were VEEV-specific RT-PCRs, seven pan-alphavirus RT-PCRs in combination with sequencing, one WTA and one HTS. In total the 13 different methods yielded 21 result entries for the WEEV sample by 16 laboratories (Table 2 and S2 Table).

The three VEEV-specific RT-PCRs all provided correct results. Pan-alphavirus assay with subsequent sequencing represented the most common strategy for VEEV testing (n = 13 entries) and had poor performance in terms of sensitivity: seven false negative results. In particular, one assay [53] proved to be unsuitable for VEEV identification since 3/3 laboratories using it failed to detect the VEEV sample (Table 2). Variable performances were observed among laboratories using the same assay, the method by Sánchez-Seco [54] was used by four laboratories and provided two correct identifications of BFV-positive sample, two false-negative results and one false-positive (identifying BFV incorrectly) on a negative control (Table 2).

**Eastern equine encephalitis virus.** Overall, 16 laboratories in nine countries tested 15 different methods to detect EEEV-positive samples of which 14 methods to the species level. Of these, five methods were EEEV-specific RT-PCRs, seven pan-alphavirus RT-PCRs in combination with sequencing, one WTA and one HTS. In total, the 15 different methods yielded 23 result entries for the WEEV sample by the 16 laboratories (Table 2 and S2 Table). As described for VEEV and WEEV, laboratories using species-specific assays (in-house or commercial) performed 100% correct in terms of both sensitivity and specificity. Pan-alphavirus assay with subsequent sequencing represented the most common strategy for EEEV testing (n = 13 entries) and had poor performance in terms of sensitivity: seven false negatives results (Table 2 and S2 Table). Pan-alphavirus conventional RT-PCR assays also lacked sensitivity. Among the 23 result entries, six false negative and one false positive result were reported. Variable performances were observed among laboratories using the same assay, Sánchez-Seco was used by four laboratories and provided two correct identifications of BFV-positive sample, two false-negative results and one false-positive (identifying BFV incorrectly) on a negative control (Table 2 and S2 Table) [54].

**Barmah forest virus.** Overall, 14 laboratories in eight countries tested 12 different methods that should be able to detect BFV-positive samples. Of these, 11 methods should be able to identify BFV-positive samples at the species level of which three methods were BFV-specific RT-PCRs, six were conventional pan-alphavirus RT-PCRs in combination with sequencing, one was WTA and one was HTS (Table 2 and S2 Table). In total 18 result entries by the 14 laboratories were submitted for the BFV sample. Laboratories mainly experienced lack of sensitivity (three false negative in 11 entries results) when using conventional pan-alphavirus RT-PCR assays in combination with sequencing, while the four real-time assays, both species-specific (n = 3) and pan-alphavirus (n = 1), provided 100% correct results. Variable performances were observed among laboratories using the same assay, e.g., the method by Sánchez-Seco was used by four laboratories and provided two correct identifications of BFV-positive sample, two false-negative results and one false-positive (identifying BFV incorrectly) on a negative control [54].

**Mayaro virus.** Overall, 15 laboratories in 10 countries tested 17 different methods to detect MAYV-positive samples. Of these, 16 methods should do so at the virus species level of which eight MAYV-specific RT-PCRs, six conventional pan-alphavirus RT-PCRs in combination with sequencing, one WTA and one HTS. In total 23 result entries by 15 laboratories were submitted for the MAYV sample (Table 2 and S2 Table).

After CHIKV, MAYV had the highest number of different species-specific RT-PCR tests that were used among the participants; despite different target regions, all assays had 100%

sensitivity and specificity for MAYV RNA. One self-developed in-house multiplex method for CHIKV and MAYV RNA detection missed the CHIKV-WA sample (Table 2 and S2 Table).

The conventional pan-alphavirus RT-PCR assays in combination with sequencing showed robust performances in MAYV detection, with one inconclusive result, one false positive result (misidentified as ONNV) and one false negative result.

**O'nyong-nyong virus.** Overall, 18 laboratories in 10 countries tested 16 different methods to detect ONNV-positive samples. Of these, 15 methods should do so at the virus species level of which seven ONNV-specific RT-PCRs, six conventional pan-alphavirus RT-PCRs in combination with sequencing, one WTA and one HTS. In total 23 result entries by 18 laboratories were submitted for the ONNV sample (Table 2 and S2 Table).

Of the seven ONNV species-specific RT-PCRs, two provided a false-negative result of which one assay did perform correctly in the hand of a different laboratory. Among the six pan-alphavirus conventional RT-PCR assays that were combined with sequencing, one laboratory mis-identified the ONNV sample as CHIKV positive which is indicative of a problem in the quality of the workflow rather than a technical problem.

HTS (1 laboratory) as primary diagnostic tool detected all panel entries correctly. WTA (1 laboratory) as primary tool had one false positive CHIKV results for one of the negative controls.

## Assays performances by method

Since RT-PCR assays represented the most common diagnostic strategy among participant laboratories, we compared outcomes of species-specific RT-PCR and pan-alphavirus RT-PCR (regardless of real-time or conventional design) (Table 2). We found that real-time tests provided a higher proportion of correct results than conventional PCR assays (96,7% *vs* 82,1%, $p < .05$). There was no statistically significant difference in performances of commercial *vs* in house RT-PCR assays, and species-specific assays vs pan-alphavirus assays ($p > .05$).

Approaches other than PCR proved to be very sensitive and quite specific. One laboratory used HTS as routine strategy in parallel with a pan-alphavirus assay [53], with 15/15 samples identified correctly (Table 2). The use of HTS allowed the laboratory to compensate a false-negative result on the VEEV sample provided by the pan-alphavirus assay they used in parallel [53]. WTA also performed well (100% sensitivity), and only one false-positive result was observed in a negative control (Table 2).

## Discussion

Twenty-three laboratories in 16 countries (15 EU/EEA, 1 EU-enlargement) participated in this EQA exercise focussing on the molecular detection of nine virus species belonging to the genus *Alphavirus*, family *Togaviridae*, i.e., SINV, CHIKV, MAYV, RRV, BFV, EEEV, VEEV, WEEV and ONNV. Reliable testing capacities for these alphaviruses in European laboratories are needed for surveillance, to diagnose illness and inform patient care, and to curb ongoing local *Aedes albopictus*-borne transmission in the case of CHIKV. Observing the capacities reported in this study we conclude that among the participating laboratories (n = 23) the capacity for CHIKV detection is perfect, i.e., 100%. Remarkably, the capacity for SINV detection was lower with 16 participating laboratories in nine countries having testing capacity. Seeing SINV endemicity in Europe it would be beneficial for patient care, surveillance, and risk assessment purposes to have a higher capacity. Based on self-reported capacities by EVD-LabNet members in the network directory [55], the capacity among member laboratories is slightly higher with 23 laboratories indicating capacity for molecular detection of RRV via specific RT-PCR and/or pan-alphavirus RT-PCR.

For the other seven alphaviruses (WEEV, VEEV, EEEV, RRV, BFV, MAYV, ONNV) the capacity for RNA detection among the 23 EQA participants ranged from 14 to 18 laboratories, in 8 to 10 countries. However, infections with RRV, MAYV, ONNV and EEEV have been described only incidentally in returning travellers while for WEEV, VEEV and BFV no travel-related cases have been reported so far. Therefore, one might question the need for a broad capacity for molecular detection of these seven viruses across Europe; even more so as introduction of these viruses by viraemic travellers will not pose a direct risk for ongoing local transmission as humans are considered dead-end host for all but RRV and/or the primary competent vectors might be absent. Therefore, the current EU/EEA capacity for detection of these viruses to a) provide information on their epidemiology in the countries where the infection was acquired, b) enable correct patient management and c) support risk assessment for public health threats in the EU and in EU-travellers might be considered sufficient under the assumption that this diagnostic capacity is readily available within the EU in a timely matter and with guaranteed reliability as is foreseen in the upcoming European Union Reference Laboratory for Public Health (EURL-PH) system.

To guarantee such reliable testing, clinical diagnostic laboratories, including those with public health functions, operate under ISO standards such as ISO15189:2022 and ISO17025. A prerequisite is the participation in proficiency testing and the implementation of corrective actions to improve the accuracy of testing if indicated. While capacities can be recorded as sufficient [56], the actual capability can be poor [57] and needs an independent assessment.

The overall testing capability for RRV was excellent with only an inconclusive call on one MAYV sample. The overall capability for CHIKV was good with little room for improvement, especially in terms of sensitivity, i.e., 90% for CHIKV-ESCA, 97% for CHIKV-Asian and 98% for CHIKV WA. However, specificity-wise there were several issues with CHIKV testing in the EQA, mainly with ONNV-positive samples that tested positive for CHIKV by five different methods, and negative samples that were identified as CHIKV-positive or inconclusive. Multiple false positive CHIKV results with ONNV samples were observed in a previous EQA as well [47] reflecting the close relationship between these two alphaviruses [28]. Laboratories operating these tests should be aware that their CHIKV test does detect ONNV as well and refer to additional testing or sequencing to resolve this when indicated (e.g., with travelers returning from Africa). CHIKV detection was assessed as well in EVD-LabNet (including predecessor ENIVD) proficiency panels in 2007 [48], 2014 [47] and 2023 [50]. In 2007, 31 EVD-LabNet laboratories participated and only 14 (45.2%) met all proficiency criteria (all samples with CHIKV concentration $\geq 4\times10^3$ per ml should be identified and no false positive results). The 2014 EQA included not only EVD-LabNet laboratories but was a worldwide assessment of 56 laboratories. Only 38% of the submitted data sets was 100% correct [47]. A CHIKV Asian lineage panel entry with a similar RNA load as in the current EQA showed a 100% score in 2023.

Besides CHIKV and RRV, testing capabilities were >90% for WEEV and SINV. As RRV, WEEV and SINV were part of a network proficiency panel for the first time, no comparisons can be made with previous assessment.

Testing capabilities for EEEV, VEEV, ONNV and MAYV show room for improvement and a new round of proficiency testing for these viruses would be beneficial. Also, a better understanding of the sensitivity of the capacity that is available in Europe is needed, using a range of RNA loads in the panel entries. The establishment in 2025 of a new EURL-PH, including an EURL for vector-borne viral diseases, could further support this type of capability building and ensure access within the EU to reliable capacity for molecular testing for these alphaviruses.

When looking at the methods used, RT-PCR represents the most used technique. Participants using species-specific formats were more likely to correctly identify samples in the proficiency panel, than laboratories using pan-alphavirus assays. Several pan-alphavirus assays proved to have low sensitivity and specificity. The lesser performance of pan-alphavirus tests appeared not only due to technical aspects of the RT-PCR test as such, as one assay that was used in combination with sequencing [54] performed 100% correct when used by some laboratories while providing false negative and/or false positive results when used by other laboratories. In this case false negative results in some laboratories could indeed be a result of technical aspects, e.g., less efficient nucleic acid extraction in the diagnostic workflow, while false positive results must be due to failure of certain quality aspects of the workstream (sample mix up, contamination, wrong reporting). In general, the lesser sensitivity of pan-alphavirus RT-PCR tests and the need for subsequent sequencing to identify the virus species, make them less suitable for first-line diagnosis.

This EQA shows very clearly that there is a lack of commercial tests available for detection of alphaviruses other than CHIKV. Besides a range of commercial tests for detection of CHIKV there was only one other commercial test used, i.e., a commercial kit for multiplex detection of the three equine encephalitis viruses (WEEV, EEEV, VEEV). Large CHIKV epidemics have encouraged manufacturers to develop and certify molecular tests but for other alphaviruses the commercial diagnostic market is most likely not large enough to develop and certify tests. The lack of commercial test development for alphaviruses might be exacerbated by the EU Regulation 2017/746 on in vitro diagnostic medical devices (IVD-R) that became effective in May 2022. The IVD-R requires manufacturers to provide large dossiers on a range of test parameters to be assessed by notified bodies. This makes it unattractive for manufacturers to start these extensive procedures for alphaviruses that do not represent an large market. As a result, molecular detection of alphaviruses other than CHIKV almost completely rely on in-house developed tests. This, together with the fact that for some alphaviruses, like RRV and BFV, very few in-house species-specific tests have been published make the overall diagnostic landscape for alphavirus testing non-uniform and not robust. Laboratory networks like EVD-LabNet and the future network of EURL-PHs can improve this situation by the organisation of proficiency testing, advise on reliable testing strategies and the promotion of standardization of testing across laboratories and countries.

This European alphavirus EQA has its limitations. The EQA did not contain any challenging samples with very low viral concentrations. Although viral loads are typically high for some alphaviruses such as CHIKV, they can vary according to the phase of the disease, nature of the sample (i.e., blood, urine, saliva) and sample handling conditions. It is advised to address sensitivity of detection in more detail in future EQAs. Furthermore, this EQA assessed only molecular diagnosis capacity, which is typically performed in the first 5-7 days post symptoms onset. Assessing performances of serology tools is fundamental to fully evaluate diagnostic capacity for alphaviruses at the European level. However, serology EQA panels for alphaviruses are a more challenging, more expensive problem due to the need for positive (virus, IgG, and/or IgM) serum.

Finally, when the EQA participants were asked whether they had implemented any corrective actions in the year following the feedback of their EQA results, about a quarter of the participants indicated that they had increased their diagnostic repertoire by adding additional assay(s). Half of the participants also expressed interest in a follow-up alphaviruses EQA. This again shows the value for laboratories of EQA of diagnostic workflows by proficiency testing and for (public) health systems in general.

## Materials and methods

### Participants

In November 2021, member laboratories of EVD-LabNet [55] were invited to register for the EQA on molecular detection of alphaviruses or to forward the invitation to competent laboratories in their respective countries. In total 65 laboratories were invited. Twenty-eight laboratories registered to participate and received EQA panels in March 2022.

### Panel composition

EQA panels were designed for molecular detection of a range of nine different alphaviruses with importance to human health, including one endemic virus (SINV), and one virus with proven transient autochthonous transmission (CHIKV) in Europe. Furthermore, the panels included seven viruses with risk of exposure for travellers outside Europe, i.e., ONNV [30], MAYV [32,58,59], BFV [60], RRV [31,61,62], EEEV [34], WEEV, VEEV. Given the history of emergence of local CHIKV cases in Europe, representative strains for all three major CHIKV lineages were included: Asian, East-Central-South African (ECSA) lineage and West-African (WA) lineage [63]. The panel consisted of 15 lyophilized, anonymized but individually coded samples: 11 plasma samples spiked with RNA of one of the nine alphavirus species and four alphavirus RNA-negative plasma samples (Table 3). WEEV, EEEV and VEEV stocks were provided by the European Union reference laboratory for equine diseases at the French Agency for Food, Environmental and Occupational Health and Safety (ANSES). Seven of the 11 viral strains used in these panels were provided by the European Virus Archive (Table 3) [64].

**Table 3. Composition of panels for molecular detection of alphaviruses, EVD-LabNet EQA, 2022.**

| EQA sample | Sample abbr. | Virus strain | RNA copies/μL | EVAg Reference |
|---|---|---|---|---|
| Chikungunya virus | CHIKV-ECSA | UVE/CHIKV/2006/RE/LR2006_OPY1 | 1.53E+04 | 001v-EVA83 |
| Chikungunya virus | CHIKV-Asian | H20235/STMARTIN/2013 | 3.83E+04 | 001v-EVA1540 |
| Chikungunya virus | CHIKV-WA | UVE/CHIKV/1983/SN/WA 37997 | 2.61E+03 | NA |
| Sindbis virus | SINV | UVE/SINV/2017/DZ/P29 | 1.41E+06 | 001V-04477 |
| O'nyong-nyong virus | ONNV | UVE/ONNV/UNK/SN/Dakar 234 | 1.73E+05 | 001v-EVA1044 |
| Mayaro virus | MAYV | UVE/MAYV/1954/TT/TC625 | 9.45E+05 | 001v-EVA502 |
| Barmah Forest virus | BFV | UVE/BFV/UNK/XX | 9.40E+05 | 001V-02828 |
| Ross River virus | RRV | T-48 | 9.60E+05 | NA |
| Eastern equine encephalitis virus | EEEV | H178/99 | 3.89E+04 | NA |
| Western equine encephalitis virus | WEEV | H160/99 | 2.87E+03 | NA |
| Venezuelan equine encephalitis virus | VEEV | UVE/VEEV/UNK/XX/TC83 | 1.14E+05 | 001v-EVA1459 |
| Negative control sample (plasma) x4 | NA | NA | NA | NA |

EQA: External Quality Assessment; abbr.: abbreviation; CHIKV-ECSA: Chikungunya virus East/Central/South African lineage; CHIKV-WA: Chikungunya virus West African lineage; NA: not applicable; EVAg: European Virus Archive

## Panel preparation and validation

For the preparation of the EQA panels, Vero E6 cells were infected with one of the 11 different virus strains (Table 3). The virus culture supernatants were heat-inactivated at 60°C for one hour. Successful inactivation was confirmed by an additional passage on cells and observation of absence of cytopathic effect and stable RNA levels over five days. Qualified non-therapeutic human plasma provided by the French blood bank was spiked to prepare 0.4 mL aliquots, that were freeze-dried in glass vials and stored at -20°C until shipment.

The viral loads per reconstituted sample were quantified in reference to in-house alphavirus-specific synthetic RNA controls. A fragment containing the virus-specific TaqMan-targeted sequence and tagged at the 5'-end with the T7 promoter sequence (5'-TAATACGACT CACTATAGGG-3') was amplified by RT-PCR using an Access RT-PCR kit (Promega). The resulting PCR products were purified and transcribed using a T7 Megashortscript kit (ThermoFisher Scientific). The obtained RNA was purified with MegaClear purification kit (ThermoFisher Scientific). RNA concentration was measured using a Nano-Drop 1000 (ThermoFisher Scientific) and translated into copy numbers. Real-time RT-PCR was performed using GoTaq qRT-PCR Kit (Promega) on a QuantStudio 12K Flex Real-Time PCR System (ThermoFisher Scientific).

The panel was further validated (pre-testing) by laboratory staff at Unité des Virus Émergents (Aix-Marseille University) who were not involved in the preparation of the EQA panels, using in-house, species-specific RT-qPCR assays [65–67]. The freeze-dried panels were shipped to all participants at ambient temperature.

## Data collection and result evaluation

We collected the EQA results for a six-week period until mid-May 2022 via an online submission form (EU-survey). Participants were asked to submit methodological information, outcomes of each assay used for their diagnosis, and their final (diagnostic) conclusion for each of the samples in the panel. Additionally, participants were invited to submit information on corrective actions taken upon the feedback of their EQA results from March to April 2023.

Submitted results were analyzed using Microsoft Excel Office 365. Statistical analyses were performed using IBM-SPSS Statistics v 24.0. Maps were created via ECDC Map Maker tool (EMMa).

Result evaluation took into account the individual testing capacity of each laboratory, i.e., the absence of various species-specific tests for a subset of the laboratories (Table 1). All laboratories had to determine the absence of any alphavirus RNA for the negative control samples. Any reported virus for the negative control samples was scored as '*False positive*'. Laboratories using test methods enabling them to discriminate between all nine different alphavirus species, had to identify the specific virus species for each alphavirus RNA containing sample. If a viral RNA containing sample was mislabelled as negative for alphavirus RNA, it was categorized as '*False negative*'. If the virus species of a RNA containing sample was mislabelled, it was categorized as '*False positive*'. The same criteria applied to laboratories with reduced testing capacity. However, this only pertained to virus RNA-containing samples for which these laboratories had species-specific tests available. For virus RNA containing samples for which the laboratory did not have reported capacity, it was sufficient to either indicate presence of unspecified alphavirus RNA (e.g., by using a pan-alphavirus test method) or absence of alphavirus RNA to pass. If an incorrect specific alphavirus species was reported for these samples, it was categorized as '*False positive*'. Any pan-alphavirus testing method that was combined with sequencing to determine the alphavirus species, was considered a species-specific test method. For any of the EQA samples laboratories also had the option to indicate as their final conclusion that their results were inconclusive.

## Supporting information

**S1 Table. Complete results table with de-identified laboratories.**
(XLSX)

**S2 Table. Performance of all assays used in the 2022 EVD-LabNet EQA on molecular detection of alphaviruses.**
(DOCX)

## Acknowledgements

We thank all the EQA participants: Institute of Public Health, Tirana, Albania; Medical University of Vienna, Center for Virology, Vienna, Austria; Institute of Tropical Medicine, Antwerp, Belgium; University Hospital for Infectious Diseases "Dr. Fran Mihaljević", Zagreb, Croatia; Cyprus Institute of Neurology and Genetics, Nicosia, Cyprus; Statens Serum Institut, Copenhagen, Denmark; Institute of Virology, Charité - Universitätsmedizin Berlin, Berlin, Germany; Bernhard Nocht Institute for Tropical Medicine, Hamburg, Germany; Friedrich-Loeffler-Institut, Federal Research Institute for Animal Helath, Greifswald - Insel Riems, Germany; Aristotle University of Thessaloniki, Thessaloniki, Greece; Molecular virology Laboratory, Microbiology and Virology department, IRCCS Fondazione Policlinico San Matteo, Pavia, Italy; Istituto Superiore di Sanità, Rome, Italy; Scientific Department, Army Medical Center, Rome, Italy; Amedeo di Savoia Hospital - ASL Città di Torino, Torino, Italy; Padova University Hospital, Padova, Italy; National Institute for Infectious Diseases Lazzaro Spallanzani IRCCS, Rome, Italy; U.O. Microbiology (CRREM), Az. Ospedaliero-Universitaria di Bologna (IRCCS), Policlinico S.Orsola-Malpighi, Bologna, Italy; Laboratoire National de Santé, Dudelange, Luxembourg; National Institute for Public Health and the Environment - RIVM, Bilthoven, Netherlands; National Institute of Health, Águas de Moura, Portugal; Institute of Microbiology and Immunology, Faculty of Medicine, University of Ljubljana, Ljubljana, Slovenia; Hospital Clinic de Barcelona, Barcelona, Spain; The Public Health Agency of Sweden - Folkhälsomyndigheten, Stockholm, Sweden; Centre National de Reference (CNR) Arbovirus, Marseille, France.

In addition, we would like to thank the ECDC support staff, as well as all the staff members who provided laboratory, administrative and other technical support at AMU (Camille Placidi, Pierre Combe and Soline Buisine) and RIVM.

## Author contributions

**Conceptualization:** Celine M. Gossner, Rémi N. Charrel, Chantal B.E.M. Reusken.

**Formal analysis:** Laura Pezzi, Ramona Moegling, Kamelia R. Stanoeva, Lance D. Presser, Willem M.R. van den Akker.

**Methodology:** Laura Pezzi, Ramona Moegling, Kamelia R. Stanoeva, Lance D. Presser, Rémi N. Charrel, Chantal B.E.M. Reusken.

**Resources:** Stephan Zientara, Celine M. Gossner, Rémi N. Charrel, Chantal B.E.M. Reusken.

**Supervision:** Rémi N. Charrel, Chantal B.E.M. Reusken.

**Validation:** Laura Pezzi, Ramona Moegling, Cecile Baronti, Pauline Jourdan, Nazli Ayhan.

**Visualization:** Laura Pezzi, Ramona Moegling, Kamelia R. Stanoeva.

**Writing – original draft:** Laura Pezzi, Ramona Moegling, Kamelia R. Stanoeva, Lance D. Presser, Willem M.R. van den Akker, Rémi N. Charrel, Chantal B.E.M. Reusken.

**Writing – review & editing:** Laura Pezzi, Ramona Moegling, Cecile Baronti, Kamelia R. Stanoeva, Lance D. Presser, Pauline Jourdan, Nazli Ayhan, Willem M.R. van den Akker, Stephan Zientara, Celine M. Gossner, Rémi N. Charrel, Chantal B.E.M. Reusken.

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
