## [Decision Letter · Decision Letter 0]

21 Nov 2024

PONE-D-24-47893Low capacity for molecular detection of Alphaviruses other than Chikungunya virus in 24 European laboratories, March 2022PLOS ONE

Dear Dr. Presser,

Thank you for submitting your manuscript to PLOS ONE. After careful consideration, we feel that it has merit but does not fully meet PLOS ONE’s publication criteria as it currently stands. Therefore, we invite you to submit a revised version of the manuscript that addresses the points raised during the review process.

We look forward to receiving your revised manuscript.

Kind regards,

Shih-Chao Lin, Ph.D.

Academic Editor

PLOS ONE

2. Thank you for stating the following financial disclosure:  [This work was financed by the European Centre for Disease Prevention and Control (ECDC) under specific contract No 1 ECD.11886 ID 23729 implementing Framework contract No ECDC/2020/010. For providing access to the virus strains used, we thank the EVA-GLOBAL consortium (funded by the European Union’s Horizon 2020 research and innovation programme under grant agreement No. 871029). Except VEEV, WEEV and EEEV, the material was provided by the European virus archive-Marseille (EVAM) under the label technological platforms of Aix-Marseille University. CBEMR and RNC were both coordinators or work package leaders for ECDC and EU Horizon funding.]. 

Additional Editor Comments:

Dear Authors,

We are informing you that your manuscript has been reviewed and has received conflicting comments, as detailed in the email. My decision regarding this submission is to request a major revision. Please revise the manuscript according to the suggestions provided by all reviewers before it can be considered further for acceptance.

Reviewers' comments:

Reviewer's Responses to Questions

**Comments to the Author**

1. Is the manuscript technically sound, and do the data support the conclusions?

Reviewer #1: Yes

Reviewer #2: Yes

Reviewer #3: Partly

2. Has the statistical analysis been performed appropriately and rigorously? 

Reviewer #1: Yes

Reviewer #2: Yes

Reviewer #3: N/A

3. Have the authors made all data underlying the findings in their manuscript fully available?

Reviewer #1: Yes

Reviewer #2: Yes

Reviewer #3: Yes

4. Is the manuscript presented in an intelligible fashion and written in standard English?

Reviewer #1: Yes

Reviewer #2: Yes

Reviewer #3: Yes

5. Review Comments to the Author

Reviewer #1: Submission well written containing all the fine details needed. The only issue is that there are some few misses that can be addressed as soon as the comments is released. Kindly address the comments as soon as possible. I enjoyed reviewing this paper because the English was not so difficult to comprehend therefor, anyone from varying background can read it.

Reviewer #2: The manuscript evaluates the molecular detection capabilities of alphaviruses in expert laboratories across Europe through an external quality assessment (EQA) in March 2022. The study involved 24 laboratories from 16 countries and used a panel of 15 samples, including nine alphaviruses and four negative controls. The study found that only CHIKV and SINV are currently circulating in Europe. While molecular detection for these was satisfactory, the overall performance for the panel was below 50%. The authors recommend improving diagnostics for alphaviruses in Europe and conducting regular EQAs to enhance laboratory capabilities, especially with the potential impact of climate change and vector expansion.

Although the research is interesting, the study acknowledges limitations that may affect the generalizability of the results. Addressing these issues would enhance the study's relevance and suitability for publication.

- The study mentions 24 laboratories as participants in the external quality assessment (EQA), but the results appear to reflect data from only 23 laboratories (24 laboratories registered for the assessment, only 23 contributed valid results). I recommend revising the title and abstract to avoid confusion and enhance clarity, ensuring the representation of the participant count is accurate.

- Could you clarify the sample types used in the study (e.g., blood, urine, saliva)? This information would help to better understand the methodology.

- The EQA did not include a wide range of viral loads or challenging samples, it may not fully represent the diagnostic capabilities of laboratories under various real-world conditions.

- The study primarily assessed molecular diagnostic methods, which may overlook the performance of serological tests. I recommend that the authors discuss this limitation.

Reviewer #3: The study “Low capacity for molecular detection of Alphaviruses other than chikungunya virus in 24 European laboratories, March 2022” by Pezzi et al. aims to address an important concern to prevent the emergence of alphaviruses in the EU through regular assessment using EQA. The manuscript is well written, but has certain limitations, addressing which would make the paper more impactful.

a) Lines 51-52 & 139-141: Authors directly mention using EQA as the diagnostic technique, without giving an introduction and pros and cons of other diagnostic methods used in the field.

b) Lines 56-60: Authors mention 15 samples, but have listed 9 alphaviruses & 4 negative controls; what were the other 2 samples?

c) The selection of 24 laboratories in 16 European countries seems to aggregate to one particular region. Did the authors have a selection criterion for choosing these laboratories? Although the authors mention that not all the laboratories invited for the study agreed to participate, approaches to include more laboratories over a wider geographical range would be a better representation for the assessment techniques.

d) Line 131: Authors state that the vector competence of competent vectors for alphaviruses other than CHIKV is not yet known. However, MAYV circulation has been observed in Haiti (Emergence of recombinant Mayaro virus strains from the Amazon basin | Scientific Reports) and Aedes aegypti has been shown to efficiently transmit MAYV under experimental conditions (Dynamic of Mayaro Virus Transmission in Aedes aegypti, Culex quinquefasciatus Mosquitoes, and a Mice Model). Therefore, these considerations should be kept in mind for future EQAs.

e) This EQA was based on detection of viral RNA, however the presence of neutralizing antibodies would be a much better measure of viral infection and would be specific to the viruses as well. Future EQAs should focus on including this approach.

6. PLOS authors have the option to publish the peer review history of their article (what does this mean? ). If published, this will include your full peer review and any attached files.

**Do you want your identity to be public for this peer review?** For information about this choice, including consent withdrawal, please see our Privacy Policy .

Reviewer #1: No

Reviewer #2: No

Reviewer #3: **Yes: ** Pallavi Rai

---

## [Author Response · Author response to Decision Letter 1]

7 Jan 2025

Thank you to all the reviewers for your reviews and comments.

---

## [Decision Letter · Decision Letter 1]

20 Jan 2025

Low capacity for molecular detection of Alphaviruses other than Chikungunya virus in 23 European laboratories, March 2022

PONE-D-24-47893R1

Dear Dr. Presser,

We’re pleased to inform you that your manuscript has been judged scientifically suitable for publication and will be formally accepted for publication once it meets all outstanding technical requirements.

Kind regards,

Shih-Chao Lin, Ph.D.

Academic Editor

PLOS ONE

Reviewers' comments:

Reviewer's Responses to Questions

**Comments to the Author**

1. If the authors have adequately addressed your comments raised in a previous round of review and you feel that this manuscript is now acceptable for publication, you may indicate that here to bypass the “Comments to the Author” section, enter your conflict of interest statement in the “Confidential to Editor” section, and submit your "Accept" recommendation.

Reviewer #3: All comments have been addressed

2. Is the manuscript technically sound, and do the data support the conclusions?

Reviewer #3: Yes

3. Has the statistical analysis been performed appropriately and rigorously? 

Reviewer #3: N/A

4. Have the authors made all data underlying the findings in their manuscript fully available?

Reviewer #3: Yes

5. Is the manuscript presented in an intelligible fashion and written in standard English?

Reviewer #3: Yes

---

## [Editor Report · Acceptance letter]

PONE-D-24-47893R1

PLOS ONE

Dear Dr. Presser,

I'm pleased to inform you that your manuscript has been deemed suitable for publication in PLOS ONE. Congratulations! Your manuscript is now being handed over to our production team.

Kind regards,

on behalf of

Dr. Shih-Chao Lin

Academic Editor

PLOS ONE